# LEARNING HIGHER-ORDER DYNAMICS IN VIDEO-BASED CARDIAC MEASUREMENT

## ABSTRACT

Computer vision methods typically optimize for first-order dynamics (e.g., optical flow). However, in many cases the properties of interest are subtle variations in higher-order changes, such as acceleration. This is true in the cardiac pulse, where the second derivative can be used as an indicator of blood pressure and arterial disease. Recent developments in camera-based vital sign measurement have shown that cardiac measurements can be recovered with impressive accuracy from videos; however, the majority of research has focused on extracting summary statistics such as heart rate. Less emphasis has been put on the accuracy of waveform morphology that is necessary for many clinically impactful scenarios. In this work, we provide evidence that higher-order dynamics are better estimated by neural models when explicitly optimized for in the loss function. Furthermore, adding second-derivative inputs also improves performance when estimating second-order dynamics. By incorporating the second derivative of both the input frames and the target vital sign signals into the training procedure, our model is better able to estimate left ventricle ejection time (LVET) intervals.

## 1 INTRODUCTION

Many of the properties of dynamical systems only become apparent when they move or change as the result of forces applied to them. In most applications we are interested in behavior in terms of positions, velocities, and accelerations, and in some cases the properties of interest may only be observed in subtle variations in the higher-order dynamics (e.g., acceleration). Whether monitoring the flight of a drone to create a control mechanism for stabilization or analyzing the fluid dynamics of the cardiovascular system in the human body, there can be a need to recover these dynamics accurately. However, most video-based systems are trained on lower-order signals, such as position in the case of landmark tracking or velocity/rate-of-change (optical flow) in the case of visual odometry (Nister et al., 2004). Thus, they optimize for lower (zeroth or first) order dynamics. Does this harm their ability to estimate higher order changes? We hypothesize that networks trained to predict temporal signals will benefit from combined multi-derivative learning objectives. To test this hypothesis, we explore video-based cardiac measurement as an example application with a complex dynamical system (the cardiovascular system) and introduce simple but effective changes to the inputs and outputs to significantly improve the measurement of clinically relevant parameters.

Photoplethysmography (PPG) is a low-cost and non-invasive method for measuring the cardiovascular blood volume pulse (BVP). There are many clinical applications for PPG as the signal contains substantial information about health state and risk of cardiovascular diseases (Elgendi et al., 2019; Reisner et al., 2008; Pereira et al., 2020). In the current world, an acutely relevant application of PPG is for pulse oximetry (i.e. measuring pulse rate and blood oxygen saturation) as it can be used to detect low blood oxygen levels associated with the onset of COVID-19 (Greenhalgh et al., 2021). The COVID-19 pandemic has accelerated the adoption of teleheath systems (Annis et al., 2020) with more and more clinical consultations being conducted virtually. Therefore, techniques for remotely monitoring physiological vital signs are becoming increasingly important (Gawałko et al., 2021; Rohmetra et al., 2021). As one might expect, with many clinical applications the precision with which the PPG signal can be recovered is of critical importance when it comes to accurate inference of downstream conditions and the confidence of practitioners in the technology.

To date, in video-based PPG measurement the primary focus of analysis and evaluation has been on features extracted from the raw waveform or its first derivative (Chen & McDuff, 2018; Liu et al., 2020; 2021; Poh et al., 2010a). However, the second derivative of the PPG signal highlights subtle features that can be difficult to discern from those in the lower derivatives. Since the second derivative reflects the acceleration (Takazawa, 1993) or the rate-of rate-of change of the blood volume, it is more closely related to the change in pressure applied by the heart on blood vessels and its relation to vascular health.

An example of a particular feature accentuated in the second-derivative (i.e. acceleration) PPG is the dicrotic notch (see Fig. 1), which occurs when the heart's aortic valve closes due to the pressure gradient between the aorta and the left ventricle. The dicrotic notch may only manifest as an inflection in the raw PPG wave; however, in the second derivative this inflection is a maxima. Inoue et al. (2017) found that the second derivative of the PPG signal can be used as an indicator of arterial stiffness - which itself is an indicator of cardiac disease. Takazawa et al. (1998) evaluated the second derivative of the PPG waveform and found that its characteristic shape can be used to estimate vascular aging, which was higher in subjects with a history of diabetes mellitus, hypertension, hypercholesterolemia, and ischemic heart disease compared to age-matched subjects without.

While the second derivative of a signal can be a rich source of information, often the zeroth- or first-order dynamics are given priority. For example, Chen & McDuff (2018) observed that training video- or imaging-based PPG (iPPG) models using first-derivative (difference) frames as input with an objective function of minimizing the mean squared error between the prediction and the *first derivative* of the target BVP signal was effective. This approach was used because the authors were designing their system to measure systolic time intervals only, which are most prominent in the lower order signals. However, they did not combine this with higher-order derivatives nor did they do any systematic comparison across derivative objectives.

We argue that a model trained with an explicit second-derivative (acceleration) objective should produce feature representations that better preserve/recover these dynamics than methods that simply derive acceleration from velocity. We observe that providing the model with a second derivative input also helps the network to better predict both the first and second derivative signals.

Finally, as diverse labeled data for training supervised models for predicting dynamical signals is often difficult to come by, we build on promising work in simulation to obtain our training data. Since light is absorbed and reflected differently for different skin tones (Bent et al., 2020; Dasari et al., 2021) having a training set that represents the true diversity of the target population is crucial for sufficient generalization. Our results show that models trained with synthetic data can learn parameters that successfully generalize to real human subjects. While this is not a central focus of our paper, we believe that it presents a promising proof-of-concept for future work.

To summarize, in this paper, we 1) demonstrate that directly incorporating higher-order dynamics into the loss function improves the quality of the estimated higher-order signals in terms of waveform morphology, 2) show that adding second-derivative inputs additionally improves performance, and 3) we describe a novel deep learning architecture that incorporates the second derivative input frames and target signals and evaluate it against clinical-grade contact sensor measurements.

## 2 BACKGROUND

**Learning Higher-Order Motion from Videos.** Despite its significance in many tasks, acceleration is often not explicitly modeled in many computer vision methods. However, there is a small body of literature that has considered how to recover (Edison & Jiji, 2017) and amplify optical acceleration (Zhang et al., 2017; Takeda et al., 2018). Given that acceleration can be equally as important as position and velocity in understanding dynamical systems, we argue that this topic deserves further attention.

A particularly relevant problem to ours is identifying small changes in videos (Wu et al., 2012; Zhang et al., 2017; Chen & McDuff, 2020; Takeda et al., 2018), and specifically in acceleration in the presence of relatively large motion. As an example, in the iPPG prediction task the aim is to identify minor changes in skin coloring due to variation in blood flow patterns, while ignoring major pixel changes due to subject or background motion. One method proposed by Zhang et al. (2017) for overcoming this signal separation problem is Video Acceleration Magnification, in which

large motions are assumed to be linear on the temporal scale of small changes while small changes deviate from this linearity. An extension to this method focused on making it more robust to sudden motions (Takeda et al., 2018). In both cases, a combination of Eulerian and Lagrangian approaches was used, rather than utilizing a supervised learning paradigm. Of relevance here is also work magnifying subtle physiological changes using neural architectures (Chen & McDuff, 2020), which have been shown to effectively separate signal and noise in both the spatial and temporal domains.

Our work might be most closely related to prior research into feature descriptors for optical acceleration (Edison & Jiji, 2017). One example uses histograms of optical acceleration to effectively encode the motion information. However, this work also defined handcrafted features, rather than learning representations from data. Our work is also related conceptually to architectures such as SlowFast (Feichtenhofer et al., 2019) in that it utilizes multiple "pathways" to learn different properties of the dynamics within a video. We were inspired by this approach; however, unlike SlowFast, we focus specifically on higher-order pathways rather than slower and faster frame sequences.

**Video-based Cardiac Measurement.** Diffuse reflections from the body vary depending on how much light is absorbed in the peripheral layers of the skin and this is influenced by the volume of blood in the capillaries. Digital cameras can capture these very subtle changes in light which can then be used to recover the PPG signal (Wu et al., 2000; Takano & Ohta, 2007; Verkruysse et al., 2008; Poh et al., 2010a). The task then becomes separating pixel changes due to blood flow from those due to body motions, ambient lighting variation, and other environmental factors that we consider noise in this context. While earlier methods leveraged source separation algorithms (Wang et al., 2016), such as ICA (Poh et al., 2010a) or PCA (Lewandowska et al., 2011), neural models provide the current state-of-the-art in this domain (Chen & McDuff, 2018; Liu et al., 2020; 2021; Song et al., 2021; Lu et al., 2021). These architectures support learning spatial attention and source-specific temporal variations and separating these from various sources of noise. Typically, the input to these models are normalized video frames and the output is a 1-D time series prediction of the PPG waveform or the heart rate. A vast majority of work has evaluated these methods based errors in heart rate estimation, which considers the dominant or "systolic" frequency alone. Only a few papers have used more challenging evaluation criteria, such as the estimation of systolic to diastolic peaks (McDuff et al., 2014).

## 3 OPTICAL BASIS

We start by providing an optical basis for the measurement of the pulse wave using a camera and specifically the second derivative signal. Starting with Shafer's Dichromatic Reflection Model (DRM)(Wang et al., 2016; Chen & McDuff, 2018; Liu et al., 2020), we want to understand how higher order changes in the blood volume pulse impact pixel intensities to motivate the design of our inputs and loss function. Based on the DRM model the RGB values captured by the cameras as given by:

$$C_k(t) = I(t) \cdot (v_s(t) + v_d(t)) + v_n(t) \tag{1}$$

where $I(t)$ is the luminance intensity level, modulated by the specular reflection $v_s(t)$ and the diffuse reflection $v_d(t)$. Quantization noise of the camera sensor is captured by $v_n(t)$. $I(t)$ can be decomposed into stationary and time-varying parts $v_s(t)$ and $v_d(t)$ (Wang et al., 2016):

$$v_d(t) = u_d \cdot d_0 + u_p \cdot p(t) \tag{2}$$

where $u_d$ is the unit color vector of the skin-tissue; $d_0$ is the stationary reflection strength; $u_p$ is the relative pulsatile strengths caused by hemoglobin and melanin absorption; $p(t)$ represents the physiological changes. Let us assume for simplicity in this case that the luminance, $I$ (i.e., illumination in the video) is constant, not time varying, which is a reasonable assumption for short videos and those in which the subject can control their environment (e.g., indoors). Then differentiating twice with respect to time, $t$:

$$\frac{\partial^2 C_k(t)}{\partial t^2} = I \cdot \left( \frac{\partial^2 v_s(t)}{\partial t^2} + \frac{\partial^2 u_d}{\partial t^2} + \frac{\partial^2 u_p(t)}{\partial t^2} + \frac{\partial^2 v_n(t)}{\partial t^2} \right) \tag{3}$$

The non-time varying part $u_d \cdot d_0$ becomes zero. Thus simplifying the equation to:

$$\frac{\partial^2 \boldsymbol{C}_k(t)}{\partial t^2} = I \cdot \left( \frac{\partial^2 \boldsymbol{v}_s(t)}{\partial t^2} + \frac{\partial^2 \boldsymbol{u}_p(t)}{\partial t^2} + \frac{\partial^2 \boldsymbol{v}_n(t)}{\partial t^2} \right) \tag{4}$$

Furthermore, if specular reflections do not vary over time (e.g., if the camera and subject are stationary), the $\boldsymbol{v}_s(t)$ term will also become zero. This means that the second derivative changes in pixel intensities are a sum of second derivative changes in PPG and camera noise. With current camera technology, and little video compression, image noise is typically much smaller than the PPG signal. Therefore, we would expect the pixel changes to be dominated by second derivative variations in the blood volume pulse:

$$\frac{\partial^2 \boldsymbol{C}_k(t)}{\partial t^2} = I \cdot \frac{\partial^2 \boldsymbol{u}_p(t)}{\partial t^2} \tag{5}$$

As such, we can infer that when attempting to estimate the second derivative of the PPG signal from videos without very large motions or illumination changes, second derivative changes in the pixel space would appear helpful and that minimizing the loss between the second derivative prediction and ground truth will be the simplest learning task for the algorithm when the input is second-derivative pixel changes.

## 4 OUR MODEL

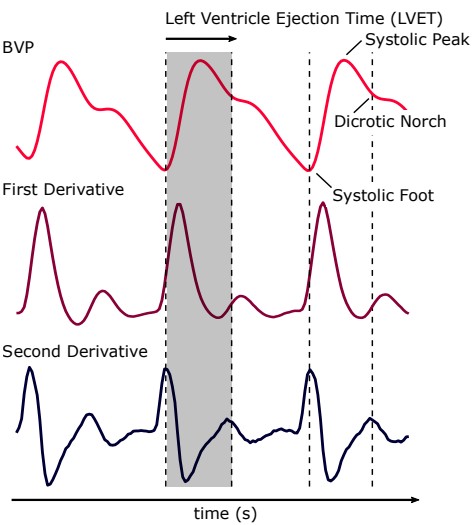

Figure 1: The Left Ventricle Ejection Time (LVET) is the duration between the beginning and end of the systolic phase. This interval corresponds to the opening and closing of the heart's aortic valve, during which the left ventricle ejects blood into the system. In the PPG waveform, this interval begins at the diastolic point and ends with the dicrotic notch.

To test our hypothesis we incorporate higher-order dynamics into a model via the loss function and as inputs to see if they will provide a better estimation of the higher-order dynamics of the target signal. We propose a multi-derivative convolutional attention network architecture (see Fig. 2) that operates separately on each set of derivative-input frames to extract sequences of features, and then maps these sequences to the target derivative signals. The convolutional attention network (CAN) paradigm was first described by Chen & McDuff (2018) to capture the spatial-temporal changes with convolutional operations and uses soft-attention masks to help segment the subject from the background and focus on relevant regions of the body. In our case, we used a CAN architecture with three branches: (1) a first-derivative branch, (2) a second-derivative branch, and (3) an attention branch. The first-derivative branch extracts features from the differences between consecutive video frames, and similarly the second-derivative branch that extracts features from the difference-of-difference frames. The attention branch uses the raw video frames to learn attention masks (one per frame) that encourage the network to prioritize regions of the image that contain useful signal (e.g. participant's skin) and ignore noisy regions (e.g. background). These attention masks are shared between the first-derivative branch and the second-derivative branch as we expect the same spatial regions to contain first and second derivative information. After feature representations are extracted from frames within each derivative-input branch, the features are concatenated together for each time step and the target signals are then generated using recurrent neural network (RNN) layers. A diagram depicting the architecture used for our experimentation is shown in Fig. 2.

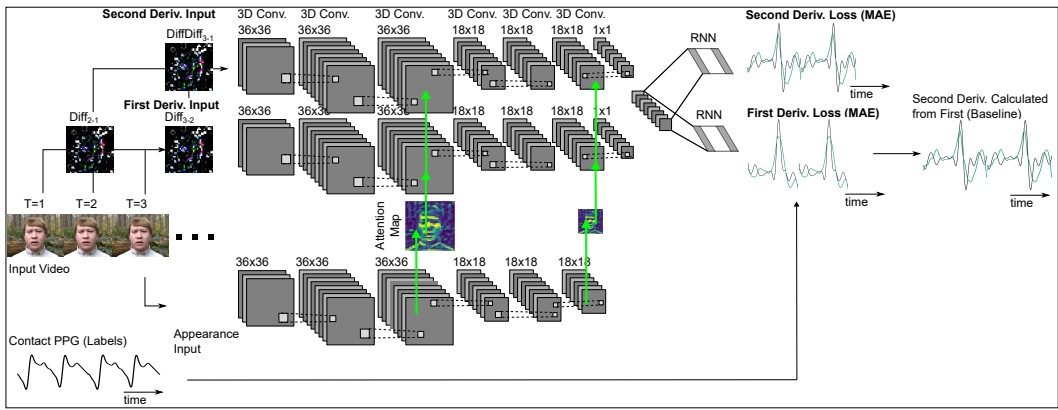

Figure 2: Our multi-derivative architecture used for experimentation. Spatial features are extracted separately for each set of derivative frames using 3D convolutional layers and mean pooling layers. Once feature representations are extracted, the temporal features from each branch are concatenated together and recurrent layers are used for modeling the temporal signals. The first- and second-derivative losses are calculated as the mean absolute error between the predictions and the synchronized ground-truth PPG signal.

## 4.1 PREDICTING MULTI-DERIVATIVE TARGET SIGNALS

The goal of iPPG is to obtain an estimate of the underlying PPG signal $p(t)$ (as in Eq. 2), while only observing video frames $X(t)$ containing a subject's skin (in this case the face). Mathematically, this can be described as learning a function: $\hat{p}(t) = f(X(t))$ or, because we are interested in changes in blood volume *changes*, estimating the first derivative of the PPG signal: $\hat{p}'(t) = f(X(t), X'(t))$, where the first derivative PPG signal is defined as: $p'(t) = p(t) - p(t-1)$. Using prior methods, to obtain an estimate of the PPG signal's second derivative, one would either differentiate the predicted PPG signal twice, or differentiate the predicted first-derivative PPG once, rather than calculate the acceleration PPG directly. In contrast, we explicitly predict the acceleration PPG waveform as a target signal. We define the second derivative waveform as the difference between consecutive first-derivative time points: $p''(t) = p'(t) - p'(t-1)$. Then we train our model to predict the second derivative waveform $\hat{p}''(t) = f(X(t), X'(t))$ given a set of input video frames $X(t)$ and the corresponding normalized difference frames $X'(t)$. To optimize our model parameters we minimize the mean squared difference between the true and predicted second derivative waveforms:

$$L = \frac{1}{T} \sum_{t=1}^{T} (p''(t) - \hat{p}''(t))^2 \tag{6}$$

## 4.2 LEVERAGING MULTI-DERIVATIVE INPUTS

It has been previously shown that the normalized difference frames are useful for predicting the first derivative PPG waveforms. Therefore, we hypothesized that incorporating the second derivative of the raw video frames $X''(t) = X'(t) - X'(t-1)$ (i.e. the difference-of-difference frames) may also be useful for predicting the PPG signal and its derivatives. Similar to the difference frames, we added a separate convolutional attention branch, where the attention mask is shared between both branches (see Fig. 2). Sharing the attention mask is a reasonable assumption as we would expect skin regions to all exhibit the signal and similar dynamics. After the feature maps in each branch are pooled into a single value per feature at each time step, the learned representations are concatenated together. These concatenated features over time are used as input sequences to the recurrent layers that generate the target waveforms.

Given that difference frames $X'(t)$ are useful for predicting the first derivative PPG waveforms, features learned from the difference-of-difference frames $X''(t)$ may be beneficial for predicting the second derivative PPG signal. In theory, if difference-of-difference features are indeed useful for predicting the acceleration PPG, then the CAN network should be able to learn those features

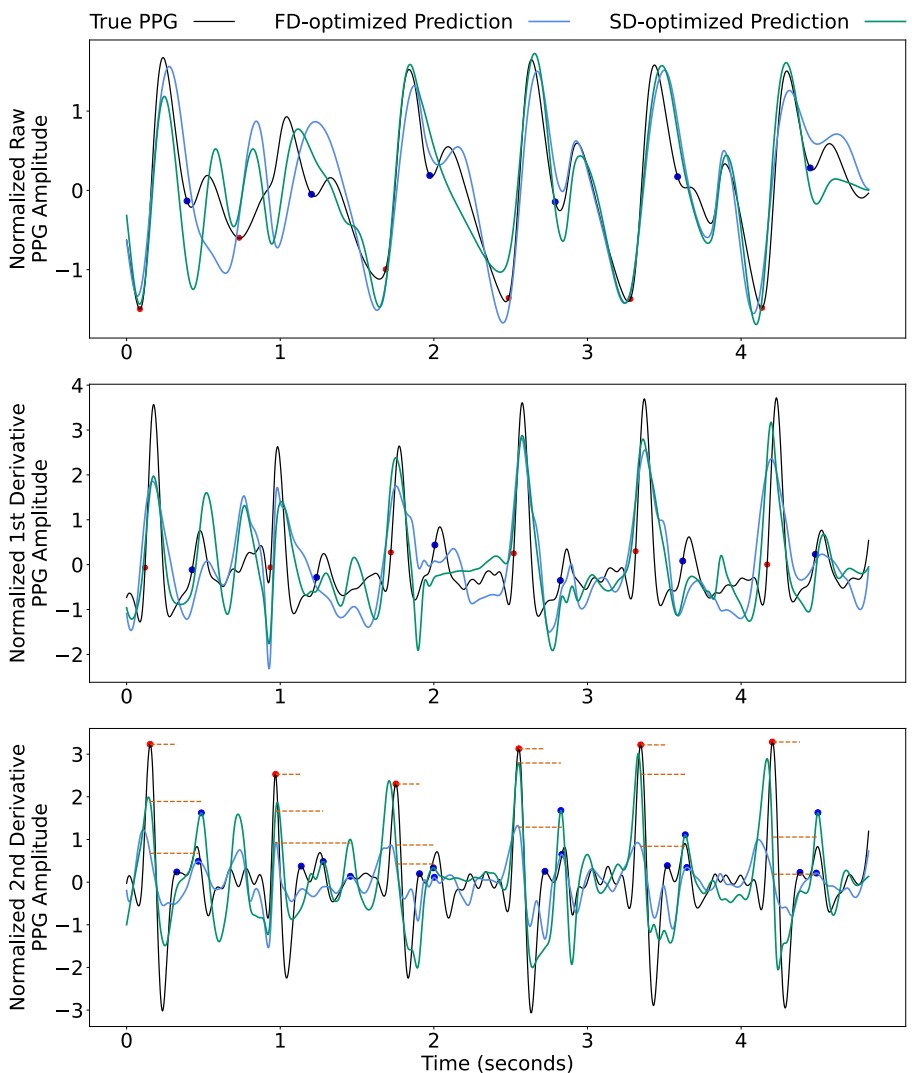

Figure 3: Comparison of true (black) and predicted (blue and green) raw or zeroth order (top), first order derivative (middle), and second order derivative (bottom) waveforms. The blue and green lines reflect two models: predicting the first derivative (blue) and predicting the second derivative (green). Diastolic points are labeled with red dots, and dicrotic notches are labeled with blue dots. LVET intervals are labeled by dotted red lines. Notice how the points of interest are generally more obvious in the second derivative waveforms, as they are maxima rather than inflections. Also note that the LVET time intervals for the second derivative model are generally more similar to those from the contact (true) PPG.

from the difference frames due to the 3D convolutional operations. However, manually adding the difference-of-difference frames could help guide the model. To examine the effect of combining higher-order inputs and target signals, we fit a model $\hat{p}''(t) = f(X(t), X'(t), X''(t))$ to predict the second-derivative PPG.

## 5 EXPERIMENTS

In this section we will describe the data used to train and evaluate our method and perform a systematic ablation study in which we test different combinations of inputs and outputs.

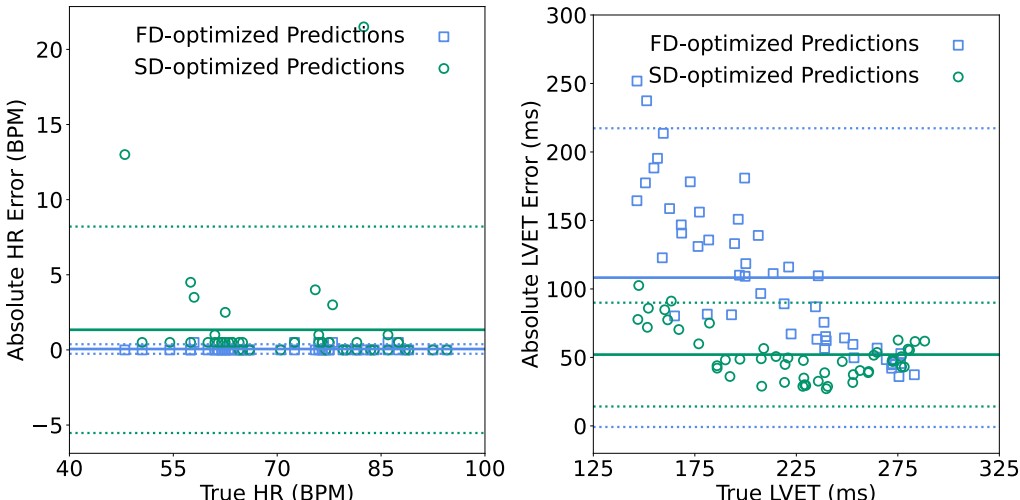

Figure 4: Bland-Altman plots comparing error distributions for average task heart rate (left) and LVET intervals (right) for the model optimized for first-derivative prediction (blue) and the model optimized for second-derivative prediction (green). (left) The absolute difference between the true and predicted heart rate for each subject/task. (right) The absolute difference between the predicted and true values (y-axis, in milliseconds) is plotted as a function of the true LVET (x-axis, in milliseconds). The solid line represents the mean error, and the dotted lines represent the 95% confidence intervals ($\pm$ 1.96 $\times$ standard deviation).

## 5.1 DATA

**Training** To train our models using a large and diverse set of subjects, we leverage recent work that uses highly-parameterized synthetic avatars to generate videos containing simulated subjects with various movements and backgrounds (McDuff et al., 2020). To drive changes in the synthetic avatars' appearance, the PPG signal is used to manipulate the base skin color and the subsurface radius (McDuff et al., 2020). The subsurface scattering is spatially weighted using an artist-created subsurface scattering radius texture that captures variations in the thickness of the skin across the face. Using physiological waveforms signals from the MIMIC Physionet (Goldberger Ary L. et al., 2000) database, we randomly sampled windows of PPG waveforms from real patients. The physiological waveform data were sampled to maximize examples from different patients. Using the synthetic avatar pipeline and MIMIC waveforms, we generated 2,800 6-second videos, where half of the videos were generated using hand-crafted facial motion/action signals, and the other half using facial motion/action signals extracted using landmark detection on real videos. Examples of the avatars can be found in Appendix A.1.1.

**Testing** Given that we are focusing on recovering *very* subtle changes in pixel intensities due to the blood volume pulse, we use a highly controlled and very accurately annotated dataset of real videos for evaluation. The AFRL dataset (Estepp et al., 2014) consists of 300 videos from 25 participants (17 male and 8 female). Each video in the dataset has a resolution of 658x492 pixels sampled at 30 Hz. Ground truth PPG signals were recorded using a contact reflective PPG sensor attached to the subject's index finger. Each participant was instructed to perform three head motion tasks including rotating the head along the horizontal axis, rotating the head along the vertical axis, and rotating the head randomly once every second to one of nine predefined locations. Since our goal in this work was to compare methods for estimating subtle waveform dynamics, which can be more difficult to do in the presence of large motion, we focused here on the first two AFRL tasks where participant motion is minimal. Examples of AFRL participants can be found in Appendix A.1.1.

Table 1: Quantitative cross-dataset performance comparison between different architecture configurations on the AFRL dataset. Models were trained using only the synthetic dataset. Values shown are (mean ± standard deviation).
Beats-per-minute (BPM); First Derivative (FD); Heart Rate (HR); Mean Absolute Error (MAE); Second Derivative (SD); Left Ventricle Ejection Time (LVET).

|  | Input Frames | | Target Signals | | HR MAE (BPM) | LVET MAE (ms) |
| --- | --- | --- | --- | --- | --- | --- |
|  | FD | SD | FD | SD |  |  |
| (FD-Optimized) | ✓ | ✗ | ✓ | ✗ | $0.66 \pm 2.07$ | $108.26 \pm 56.19$ |
|  | ✓ | ✗ | ✓ | ✓ | $1.63 \pm 4.21$ | $64.77 \pm 31.04$ |
|  | ✓ | ✗ | ✗ | ✓ | $1.68 \pm 4.85$ | $57.29 \pm 22.68$ |
|  | ✓ | ✓ | ✓ | ✗ | $0.88 \pm 2.75$ | $75.41 \pm 43.47$ |
|  | ✓ | ✓ | ✓ | ✓ | $1.16 \pm 3.67$ | $65.99 \pm 31.88$ |
| (SD-Optimized) | ✓ | ✓ | ✗ | ✓ | $3.41 \pm 8.77$ | $52.07 \pm 19.53$ |

## 5.2 Implementation details

We trained our models using a large dataset of generated synthetic avatars and evaluated model performance on the AFRL dataset, which consists of real human subjects. For each video, we first cropped the video frames so that the face was approximately centered. Next, we reduced the resolution of the video to 36x36 pixels to reduce noise and computational requirements while maintaining useful spatial signal Verkruysse et al. (2008); Wang et al. (2017); Poh et al. (2010b). The input to the attention branch was $T$ raw video frames. The input to the first-derivative branch was a set of $T$ normalized difference frames, calculated by subtracting consecutive frames and normalizing by the sum. The input to the second-derivative branch was a set of $T - 1$ difference-of-difference frames (second derivative frames), calculated by subtracting consecutive normalized difference frames (i.e. the $T$ frames used as input to the motion branch). In our experiments, we used a window size of $T = 30$ video frames to predict the target signals for the corresponding 30 time points. During training, a sliding window of 15 frames (i.e. 50% overlap between consecutive windows) was used to increase the total number of training examples. The model was implemented using Tensorflow (Abadi et al., 2016) and trained for eight epochs using the Adam (Kingma & Ba, 2017) optimizer with a learning rate of 0.001, and a batch size of 16.

## 5.3 Systematic Evaluation

To measure the effect of using multi-derivative inputs and outputs, we systematically removed the second-derivative parts of the model and used quantitative and qualitative methods to examine the change in model performance. To quantitatively measure the quality of the predicted signal, we calculated two clinically important parameters - heart rate (HR) and the left ventricular ejection time (LVET) interval (see Appendix A.1.3 for details). Video-based HR prediction has been a major focus of iPPG applications, with many methods showing highly-accurate results. HR can be determined through peak detection or by determining the dominant frequency in the signal (e.g. using fast Fourier transform). Since current iPPG methods are able to achieve sufficiently-low error rates on the HR estimation task, we believe that metrics that capture the quality of waveform morphology should also be considered.

The LVET interval is defined as the time between the opening and closing of the heart's aortic valve, i.e. the systolic phase when the heart is contracting (see Fig. 1). In the PPG waveform, this interval begins at the diastolic point (i.e. the global minimum pressure within a heartbeat cycle) and ends with the dicrotic notch (i.e. local minimum occurring after systolic peak, marking the end of the systolic phase and the beginning of the diastolic phase). LVET typically is correlated with cardiac output (stroke volume × heart rate)(Hamada et al., 1990), and has been shown to be an indicator of future heart failure as the time interval decreases with left-ventricle dysfunction (Biering-Sørensen et al., 2018).

Calculating LVET requires identification of the diastolic point and the dicrotic notch. The diastolic point is a (global) minimum point within a heart beat, meaning it corresponds to a positive peak

in the second derivative signal according to the second-derivative test. Similarly, the dicrotic notch is a (local) minimum in the PPG signal, and appears as a positive peak in the second derivative following the diastolic peak in time. Because the dicrotic notch can often be a subtle feature, it is much easier to identify in the PPG's second derivative compared to the raw signal. Therefore, it is a good example of clinically-important waveform morphology that is best captured by higher-order dynamics.

**Removing the second-derivative frames** In Table 1, quantitative evaluation metrics (HR and LVET) are shown for all experiments in our ablation study, using tasks 1 and 2 from the AFRL dataset. Removing the second-derivative (SD) frames results in the model configurations in the top three rows of Table 1. When SD frames are removed, the result is a general decrease in the HR error. However, there is also a general increase in LVET interval prediction error, which suggests that including the SD frames leads to improved estimation of waveform morphology.

**Removing the first-derivative target signal** Intuitively, models that are optimized using a loss function specifically focusing on a single objective will perform better in terms of that objective compared to models trained with loss functions containing multiple objectives. By removing the first-derivative target signal from the training objective, the model is focused to exclusively focus on the second-derivative (SD) objective. Empirically, this leads the SD-Optimized model to have the lowest LVET MAE of any model configuration (last row of Table 1). While the SD-Optimized model achieves the lowest LVET error, the HR error is the highest of any configuration. These results suggest that there are performance trade-offs to consider when designing a system for particular downstream tasks.

**Removing the second-derivative target signal** When the second-derivative target signal is removed from the model, the optimization procedure is purely focused on improving the prediction of the first derivative. The FD-Optimized model (first row of Table 1) serves as a form of baseline, since previous works have focused on using first-derivative (FD) frames to predict the first-derivative PPG signal. Fig. 4 shows a Bland-Altman plot (Martin Bland & Altman, 1986) comparing the FD-Optimized and SD-Optimized error distributions as a function of the ground-truth values both HR and LVET intervals.

Perhaps unsurprisingly, our results show the FD-Optimized model achieves the lowest HR MAE ($0.66 \pm 2.07$ BPM) of any model configuration examined and, in particular, improves HR estimation compared to models without the first derivative target signal. However, the FD-Optimized model also has the worst performance in terms of the LVET MAE ($108.26 \pm 56.19$ ms) of any model configuration. This suggests that while the configuration provides an accurate assessment of the heartbeat frequency, the quality of predicted waveform morphology can be improved by incorporating second-derivative information. We observe similar results when evaluating the models on the UBFC (Bobbia et al., 2019) and PURE (Stricker et al., 2014) datasets (see Appendix Table 3).

**Qualitative comparisons** For a qualitative comparison, in Fig. 3 we plot the ground-truth, FD-Optimized, and SD-Optimized PPG, first derivative, and second derivative. Additionally, in the bottom panel of Fig. 3 we overlay the true and predicted LVET intervals for each signal to demonstrate model performance. For additional qualitative comparisons, see Appendix A.2.

## 6 CONCLUSIONS

Using the task of video-based cardiac measurement we have shown that when learning representations for dynamical systems that appropriately designing inputs, and optimizing for derivatives of interest can make a significant difference in model performance. Specifically, there is a trade-off between optimizing for lower-order and higher-order dynamics. Given the importance of second-derivatives (i.e., acceleration) in this, and many other video understanding tasks, we believe it is important to understand the trade-off between optimizing for targets that capture different dynamic properties. In cardiac measurement in particular, the LVET is one of the more important clinical parameters and can be better estimated using higher-order information. While we have investigated the importance of higher-order dynamics in the context of video-based cardiac measurement, this paradigm is generally applicable. We believe future work will continue to showcase the importance of explicitly incorporating higher-order dynamics.

## 7 ETHICS STATEMENT

Camera-based cardiac measurement could help improve the quality of remote health care, as well as enable less invasive measurement of important physiological signals. The COVID-19 pandemic has revealed the importance of tools to support remote care. These needs are likely to be particularly acute in low-resource settings where distance, travel costs, and time are a great barrier to access quality healthcare. However, given the non-contact nature of the technology, it could also be used to measure personal data without the knowledge of the subject. Just as is the case with traditional contact sensors, it must be made transparent when these methods are being used, and subjects should be required to consent before physiological data is measured or recorded. There should be no penalty for individuals who decline to be measured. New bio-metrics laws can help protect people from unwanted physiological monitoring, or discrimination based on pre-existing health conditions detected via non-contact monitoring. However, social norms also need to be constructed around the use of this technology.

In this work, data were collected under informed consent from the participants.

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

# A  APPENDIX

## A.1  SUPPLEMENTAL METHODS

### A.1.1  EXAMPLE VIDEO FRAMES

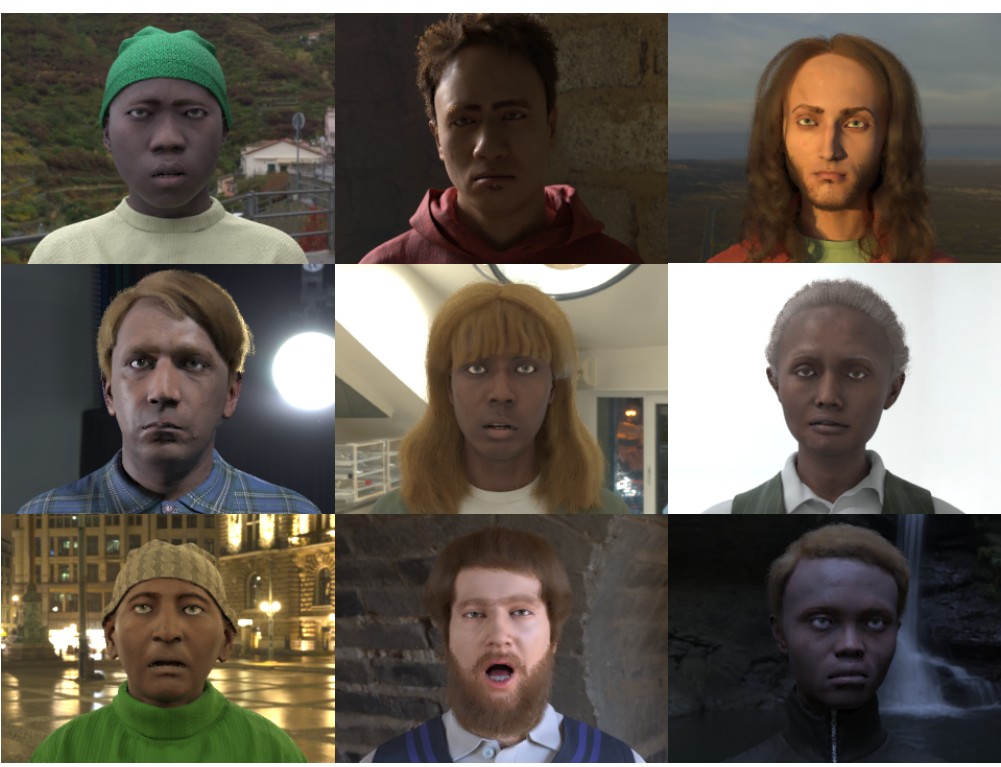

Figure 5: Example video frames of different synthetic avatars generated for the training data set. The highly-parameterized avatar generation pipeline enables the creation of diverse subjects with varied demographics, lighting conditions, backgrounds, clothing/accessories, and movements.

### A.1.2  MODEL ARCHITECTURE

The first two 3D convolutional layers in each branch each have 16 filters and the final two 3D convolutional layers in each branch each have 32 filters. Each convolutional layer has a filter size of 3x3x3 for all 3D convolutional layers in the network. All convolutional layers are padded such that they have the same height, width, and number of time steps in each consecutive layer. Convolutional layers use the hyperbolic tangent activation function, except for the convolutional layers used for the attention masks which use a sigmoid activation function for generating the soft masks. Attention masks (one per time step) are applied by applying an element-wise multiplication of the attention mask with each 3D convolutional feature map. Average pooling layers reduce the height and width of the frames by a factor of two, except for the final average pooling layer that pools over the entire frame (i.e. reduces each feature map to a single value per time step). Dropout (25% probability) is applied after every pooling layer to reduce overfitting.

After the final pooling layer, the learned features for each time step in a branch are concatenated together (i.e. combined across branches to share information). Each target signal uses its own set of (2) RNN layers to read the concatenated features over time and generate a target sequence. The first RNN layer is implemented as a bi-directional GRU (hyperbolic tangent activation function) with 64 total units (32 each direction). The second RNN layer is a GRU (linear activation function) layer with 1 output value per time step.

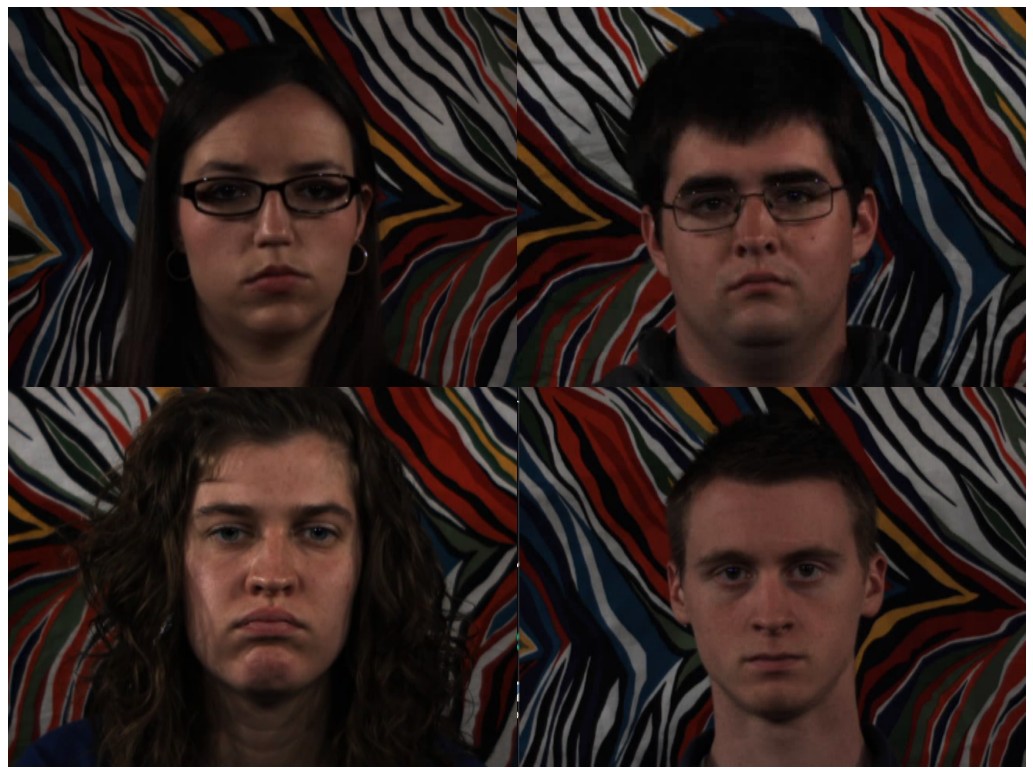

Figure 6: Example video frames from four participants in the AFRL dataset used for model testing and evaluation.

### A.1.3 METRIC CALCULATION

**Heart Rate (HR) estimation** To estimate the heart rate, we use an fast Fourier transform (FFT)-based method to calculate the dominant frequency in the signal, which corresponds to the heart rate. We first estimate power spectral density using the "periodogram" function from the scipy.signal (Virtanen et al., 2020) library. Then we band-pass filter the PPG signal, with cutoff frequencies of 0.75-4.0 Hz (corresponding to a minimum HR of 45 BPM and maximum HR of 240 BPM). Finally, we select the frequency with the maximum power, and use this as our estimated HR.

**Left Ventricle Ejection Time (LVET) estimation** The LVET time is defined as the time interval between the diastolic peak and the dicrotic notch. To calculate this interval, we first identified the diastolic point in the second derivative (SD) of the PPG signal, which, because it is a "global" minima in the PPG heartbeat, appears as a "global" maxima (positive SD value) in the SD PPG. Then, in each predicted SD PPG waveform, we identified candidate dicrotic notch points. Since the dicrotic notch manifests as a "local" minima in the PPG signal, it appears as a "local" maxima in the PPG SD signal (positive SD value). Using peak detection ("find_peaks" function in the scipy.signal library (Virtanen et al., 2020)) we identify candiadate dicrotic notch points by finding local peaks that occur after a diastolic point, and use the dicrotic notch candidate point that is closest in time to the reference diastolic point.

Because both the ground truth PPG (and therefore its derivatives) and, in particular, the predicted PPG (and its derivatives), contain signal artifacts and noise, the peak detection process is not perfect. To reduce variability in the LVET interval estimates due to noise, we apply a smoothing operation. Specifically, we estimate the mean LVET interval within a 10-second non-overlapping window and use this as our estimate of true/predicted LVET. See Appendix Fig. 7 for example LVET intervals over time, and the estimated LVET intervals after smoothing within windows.

## A.2 SUPPLEMENTAL RESULTS

Table 2: Quantitative performance comparison between different architecture configurations. Values shown are (mean ± standard deviation).
Beats-per-minute (BPM); First Derivative (FD); Heart Rate (HR); Mean Absolute Error (MAE); Second Derivative (SD); Left Ventricle Ejection Time (LVET).

| Input Frames | | Target Signals | | HR MAE (BPM) | LVET MAE (ms) |
|---|---|---|---|---|---|
| FD | SD | FD | SD | | |
| ✗ | ✓ | ✓ | ✗ | 3.14 ± 7.07 | 83.09 ± 42.41 |
| ✗ | ✓ | ✓ | ✓ | 6.97 ± 16.30 | 65.10 ± 31.56 |
| ✗ | ✓ | ✗ | ✓ | 2.95 ± 6.57 | 57.67 ± 25.82 |

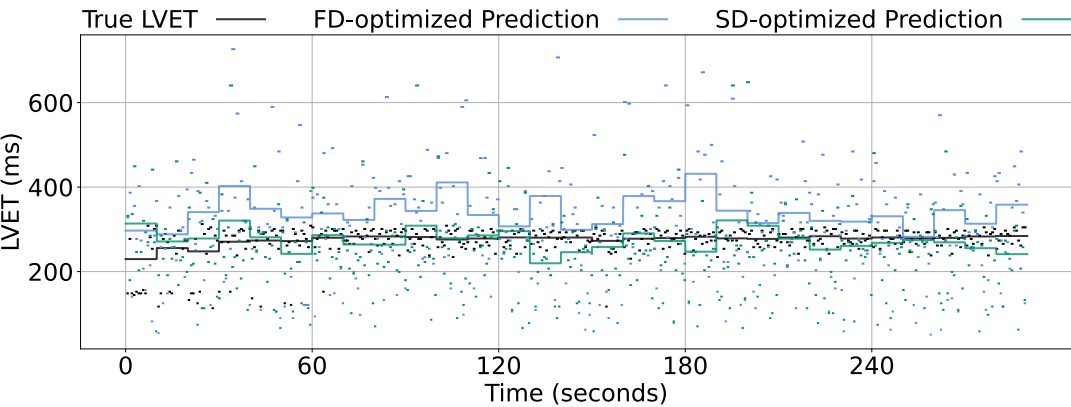

Figure 7: Comparison of Left Ventricle Ejection Time (LVET) estimation over a 5-minute time period. Solid lines are computed as the mean LVET interval within non-overlapping 10-second windows.

Table 3: Quantitative cross-dataset performance comparison between different architecture configurations in two separate external datasets: UBFC (Bobbia et al., 2019) and PURE (Stricker et al., 2014). Models were trained only on the synthetic dataset. Descriptions of the FD-optimized and SD-optimized models can be found in Table 1. Values shown are (mean ± standard deviation).
Beats-per-minute (BPM); First Derivative (FD); Heart Rate (HR); Mean Absolute Error (MAE); Second Derivative (SD); Left Ventricle Ejection Time (LVET).

| Model | UBFC | | PURE | |
|---|---|---|---|---|
| | HR MAE (BPM) | LVET MAE (ms) | HR MAE (BPM) | LVET MAE (ms) |
| (FD-Optimized) | 3.59 ± 9.84 | 80.90 ± 58.99 | 4.17 ± 2.99 | 75.84 ± 18.74 |
| (SD-Optimized) | 3.88 ± 9.85 | 41.90 ± 15.72 | 4.18 ± 3.26 | 37.85 ± 3.57 |

