# OpenReview forum: "Learning Higher-Order Dynamics in Video-Based Cardiac Measurement"
_ICLR.cc/2022/Conference — ICLR 2022 Submitted_

### Official Review · Reviewer_W7ES · 2021-10-30

**Correctness:** 3
**Technical Novelty And Significance:** 2
**Empirical Novelty And Significance:** 2
**Recommendation:** 5
**Confidence:** 2

**Main Review:**

Strengths:

-Thorough ablation study

-Considering second-order motion is a novel topic

Weakness:

-Technical novelty seems to be limited. The convolutional attention network (CAN) paradigm was introduced before and it will be better to have some discussion on what is the new component added by comparison with the original CAN.

-Further discussion is required for some results. More discussion is appreciated for the results in Table 1. Intuitively, the first-order and second-order motion is highly correlated. If the first-order performance improves, the second-order performance should also improve and vice versa. On the other hand, doing both first and second-order estimation could be viewed as multi-task learning and they should benefit from each other. However, it shows that for this task, removing first-order loss will improve second-order performance. Also, if add both losses, the second-order performance decreased. This is a kind of strange phenomenon and more discussion is appreciated.

-Lack some references in PPG/cardiac motion estimation
The strange results might be because the overfitting or generalization issues. The meta-learning has been proven to be useful for PPG estimation/cardiac motion estimation tasks. It will be interesting to add the following references and have some discussions:

Lee, Eugene, Evan Chen, and Chen-Yi Lee. "Meta-rppg: Remote heart rate estimation using a transductive meta-learner." European Conference on Computer Vision. Springer, Cham, 2020.

Yu, Hanchao, et al. "Foal: Fast online adaptive learning for cardiac motion estimation." Proceedings of the IEEE/CVF Conference on Computer Vision and Pattern Recognition. 2020.

**Summary Of The Paper:**

Estimating the motion signal from the video is an important task with applications in computer vision and healthcare. In this paper, a multi-derivative convolutional attention network is used to estimate the high order derivatives of the 1D heart beating signals, with video data as input. A second-order loss is used and the results are evaluated in first-order(HR MAE) metrics and second-order metrics (LEFT MAE). The results show the second-order loss will improve the second-order metrics while harming the first-order performance.

**Summary Of The Review:**

In summary, the higher-order motion characteristics estimation from the video is an interesting topic while I have some concerns on the novelty and the experiment results.

---

> ### Author Response · Authors · 2021-11-22
> **Reviewer Response**
>
> We thank the reviewer for providing such useful feedback and posing important questions. Below we do our best to address the insightful points raised by the reviewer:
>
> > P1. Technical novelty seems to be limited. The convolutional attention network (CAN) paradigm was introduced before and it will be better to have some discussion on what is the new component added by comparison with the original CAN.
>
> We agree with the reviewer that the novelty of the proposed architecture is limited. However, this was specifically by design, as the goal of our work was not to propose a new architecture but rather to emphasize the importance of incorporating higher-order inputs/target signals into the model/optimization procedure.
>
> > P2. Further discussion is required for some results. More discussion is appreciated for the results in Table 1. Intuitively, the first-order and second-order motion is highly correlated. If the first-order performance improves, the second-order performance should also improve and vice versa. On the other hand, doing both first and second-order estimation could be viewed as multi-task learning and they should benefit from each other. However, it shows that for this task, removing first-order loss will improve second-order performance. Also, if add both losses, the second-order performance decreased. This is a kind of strange phenomenon and more discussion is appreciated.
>
> We thank the reviewer for pointing out this important point which we were hoping to make clear to the reader. While multi-task learning is supposed to improve the average performance in cases where the tasks are related enough that they can leverage shared information, in some cases for specific tasks the multi-task model performance can be worse than the single-task model performance. This phenomenon has been previously termed “negative transfer” [1]. In Liu et al. [1], the authors propose one reason for negative transfer being that one (group of) the related tasks dominates the training process. It’s possible that the first-derivative target waveform dominates the more subtle signals in the higher-order targets. Therefore, the first-derivative prediction performance improves when adding the second-derivative as an additional target, but to obtain the best estimate of the more subtle second-derivative dynamics removing the dominating first-derivative target helps.
>
> References
>
> [1] Liu, S., Liang, Y., & Gitter, A. (2019). Loss-Balanced Task Weighting to Reduce Negative Transfer in Multi-Task Learning. Proceedings of the AAAI Conference on Artificial Intelligence, 33(01), 9977-9978.

---

> > ### Comment · Reviewer_W7ES · 2021-11-26
> > **Reviewer Response**
> >
> > 1. Since the technical novelty is limited, I am not sure if this work meets the standard of ICLR  with an emphasis on learning representation.
> > 2. As mentioned by the authors, "It’s possible that the first-derivative target waveform dominates the more subtle signals in the higher-order targets." To reduce this phenomenon, one can try to fine-tune the weights between different losses. A follow-up question will be: have you tried to change the weight between the losses?
> > 3. It seems the authors did not reply to the missing reference question

---

> > > ### Author Response · Authors · 2021-12-01
> > > **Thanks.**
> > >
> > > Thanks for your response.  We apologize for not responding to your question about the missing references.  We agree that these are relevant and will add these to the paper.  Unfortunately we cannot upload another revision at this point, but will do so when we can.
> > >
> > > Regarding your question about the loss trade-off.  We agree that this would be one way to balance or tune the first and second order waveform shapes.  We are running additional experiments with different loss weights (0.25 FO + 0.75 SO), (0.50 FO + 0.5 SO), etc. where FO = first order and SO = second order. We will report these results as soon as we have them.
> > >
> > > We would argue that our work does contribute to the understanding of learning representations, as we explore how to improve the ability for models to learn representations for high-order dynamics.

---

### Official Review · Reviewer_Z3zg · 2021-11-01

**Correctness:** 3
**Technical Novelty And Significance:** 3
**Empirical Novelty And Significance:** 3
**Recommendation:** 6
**Confidence:** 4

**Main Review:**

Strong Points:

The paper is clear and very well written. In addition, the problem is clearly stated, and the methodology is presented with the corresponding metrics.

This work has a strong background motivation and the methodology is novel, since there is currently a need to improve the models whose tasks are represented in higher dynamics, as shown in the introduction.

The paper evaluates correctly the incorporation of the second derivative of the input frames to improve the performance of second order dynamics, which enriches the results of the work.

It is very important and beneficial that the authors have shared their work on Github, so that the work is reproducible and replicable. This enriches the work presented.


Weak Points:

There is a lack of clear support for the use of a synthetic dataset instead of a real one. There are differences that need to be clearly defined and justified as to how much it affects the performance of the convolutional network.

Although the use of the attention mask is a reasonable assumption, there is a lack of comparative results of the full model with and without the attention mask. Obtaining these differences is important to understand how it affects the use of care models. The comparison is not complicated and is suggested.

The proposed architecture is interesting and works well, however there is a lack of comparison with other architectures to evidence that the proposed architecture is the best. In this way, a comparison with simpler and more complex models could be shown to verify that the implemented model is the best one.

The paper mentions that in the implementation the video resolution is reduced to 36x36 pixels. But, it is not supported how the reduction of resolution affects performance. One way to present this could be through a comparison of performance, computational power consumption, and response time at different resolutions.

The number of epochs used in the training stage is eight, which is relatively low. In addition, there is no support as to why this low number of epochs was chosen. Making use of more epochs would help a model to generalize better.


Suggestions:

In the Introduction section, you mentioned that "To date, in video-based PPG measurement the primary focus of analysis and evaluation has been on features extracted from the raw waveform or its first derivative", however, no research works are cited or referenced. The references that support this statement should be cited.

At section 4.2, the paper states: “Given that difference frames X’(t) are useful for predicting the first derivative PPG waveforms, features learned from the difference-of-difference frames X’’(t) may be beneficial for predicting the second derivative PPG signal.”  I think I should say: “Given that difference frames X(t) are useful for predicting the first derivative PPG waveforms, features learned from the difference-of-difference frames X’(t) may be beneficial for predicting the second derivative PPG signal.”

How much does the amount of light affect the prediction? It would be interesting to show a graph or table of the variation of results as the illumination is increased or decreased. Also, would it be better to perform the prediction on natural light? How would artificial light vary the results (e.g., light operating at a frequency of 60 Hz changes the illumination and the camera could capture at the moment of less illumination which would change the results)?


**Summary Of The Paper:**

Computer vision algorithms based on deep learning usually optimize first-order dynamics. But, in cases the properties of interest are small variations, which are best described in higher order, such as acceleration. A direct application of this is the heart pulse, where the second derivative can be used as an indicator of blood pressure and arterial disease. In this work, the author(s) propose a methodology for beat waveform prediction using convolutional attention networks, and  considering the derivatives of the input images and optimizing for derivatives of interest. The results are very interesting and show that by appropriately incorporating higher-order dynamics, the performance of video understanding tasks can be greatly improved.

**Summary Of The Review:**

The work presented is of general interest and contributes with a methodology for a better understanding of learning representations for dynamical systems. The paper addresses several aspects such as the optimization of the loss function of neural network models to better estimate higher order dynamics, as well as the incorporation of the second derivative of the input frames to improve the performance of second order dynamics. Applying this to the beat wave characterization problem, the model is able to better estimate left ventricular ejection time intervals (LVET). The paper will be greatly improved if the revisions suggested in this review are taken into account, especially improving the comparative results with other models and better justifying the use of synthetic data.

---

> ### Author Response · Authors · 2021-11-22
> **Reviewer Response (part 1)**
>
> We thank the reviewer for a thorough and thoughtful critique of our paper, the feedback was very helpful and we believe it has allowed us to improve the paper. Based on the feedback we have made corresponding updates to the text which hopefully clarify some of the points brought up. Below we provide our responses to the great points that the reviewer brought to our attention:
>
> > P1. There is a lack of clear support for the use of a synthetic dataset instead of a real one. There are differences that need to be clearly defined and justified as to how much it affects the performance of the convolutional network.
>
> We apologize for not being more explicit regarding the motivation behind using synthetic data instead of real data. Since light is absorbed and reflected differently for different skin tones [1,2] having a training set that represents the true diversity of the target population is crucial for sufficient generalization. Additionally, using synthetic data allows us to quickly generate large datasets that would otherwise be difficult to obtain due to the time and resources needed to conduct studies with real participants, without any privacy issues. We have updated the paper to clarify and support the use of synthetic data.
>
> > P2. Although the use of the attention mask is a reasonable assumption, there is a lack of comparative results of the full model with and without the attention mask.
>
> Prior work has shown that adding the attention mask to the network architecture does, in general, improve the results and interpretability of the network [3]. Additionally, since the motivation for our work was not to propose a new network architecture, but rather to emphasize the importance of incorporating higher-order inputs/target signals into the model/optimization procedure, we wanted to hold the network architecture as fixed as possible across experiments. Therefore, we felt that the comparison between a full model with and without the attention mask was not justified.
>
> > P3. The proposed architecture is interesting and works well, however there is a lack of comparison with other architectures to evidence that the proposed architecture is the best.
>
> Since the goal of this work was not to propose and evaluate a new rPPG neural architecture, but rather to emphasize the importance of incorporating higher-order inputs/target signals into the model/optimization procedure, we did not feel that it made sense to add additional popular estimation networks into the comparison.  Are there specific architectures that the reviewer felt would be appropriate?
>
> > P4. The paper mentions that in the implementation the video resolution is reduced to 36x36 pixels. But, it is not supported how the reduction of resolution affects performance.
>
> Previous work in rPPG has shown aggressive spatial averaging (i.e., downsampling) is helpful for boosting the PPG signal-to-noise ratio [4,5,6]. The averaging helps to reduce the effect of sensor noise which is generally assumed to be Gaussian. We leverage this principle here, and as demonstrated by the testing set performance, which uses real videos collected from real human participants, we achieve good results in HR and LVET estimation.
>
> References
>
> [1] Bent et al. Investigating sources of inaccuracy in wearable optical heart rate sensors; npj Digital Medicine (2020).
>
> [2] Dasari et al. Evaluation of biases in remote photoplethysmography methods; npj Digital Medicine (2021).
>
> [3] Chen, W., & McDuff, D. (2018). Deepphys: Video-based physiological measurement using convolutional attention networks. In Proceedings of the European Conference on Computer Vision (ECCV) (pp. 349-365).
>
> [4] Verkruysse, W., Svaasand, L. O., & Nelson, J. S. (2008). Remote plethysmographic imaging using ambient light. Optics express, 16(26), 21434-21445.
>
> [5] Poh, M. Z., McDuff, D. J., & Picard, R. W. (2010). Non-contact, automated cardiac pulse measurements using video imaging and blind source separation. Optics express, 18(10), 10762-10774.
>
> [6] Wang, W., den Brinker, A. C., Stuijk, S., & De Haan, G. (2016). Algorithmic principles of remote PPG. IEEE Transactions on Biomedical Engineering, 64(7), 1479-1491.

---

> > ### Comment · Reviewer_Z3zg · 2021-11-25
> > **Reviewer comments**
> >
> > Thank you very much for the clarifications and your responses to the review of your paper.
> >
> > As you mention in P1, synthetic data generation increases generalization because you have a greater number of foot tones, and at the same time you avoid privacy issues in data collection. My question was more about the level of detail of the biological patterns that could be generated synthetically.
> >
> > For P2 and P2, I understand that the work does not focus on evaluating the performance of the model with or without the attention technique, or to propose the best convolutional network model, but it would have helped to describe the neural network more extensively.
> >
> > The justification for point P4 is clear, and the references cited help to support it.

---

> > > ### Author Response · Authors · 2021-12-01
> > > **Thanks.**
> > >
> > > Thanks for your response. We added additional details about the model in the appendix and will further elaborate on the implementation details. Are there specific important details you feel are missing?  Our code is also provided as supplementary material for a precise implementation. At the moment we do not believe we can upload a new revision but we will do so as soon as we have the opportunity via OpenReview.

---

> > > > ### Comment · Reviewer_Z3zg · 2021-12-04
> > > > **Reply to Thanks**
> > > >
> > > > From my side I think there are no more relevant details that need to be clarified. I will review what has been added to the appendix about the details of the model and the code, as soon as they can upload a new version. Regards.

---

> ### Author Response · Authors · 2021-11-22
> **Reviewer Response (part 2)**
>
> > P5. The number of epochs used in the training stage is eight, which is relatively low. In addition, there is no support as to why this low number of epochs was chosen. Making use of more epochs would help a model to generalize better.
>
> We chose the number of epochs to be eight since we found that the network’s validation loss had converged after 8 epochs.
>
> > P6. In the Introduction section, you mentioned that "To date, in video-based PPG measurement the primary focus of analysis and evaluation has been on features extracted from the raw waveform or its first derivative", however, no research works are cited or referenced. The references that support this statement should be cited.
>
> We apologize for the lack of proper citation. We have updated the text with references that support this statement.
>
> > P7. How much does the amount of light affect the prediction? It would be interesting to show a graph or table of the variation of results as the illumination is increased or decreased. Also, would it be better to perform the prediction on natural light? How would artificial light vary the results (e.g., light operating at a frequency of 60 Hz changes the illumination and the camera could capture at the moment of less illumination which would change the results)?
>
> Several studies have characterized how the performance of camera rPPG changes with light composition and intensity [7].  Generally, broad spectrum (e.g., natural light) and higher (but not too bright) intensity illumination is optimal.  However, rPPG works with many lighting types (LED, incandescent, fluorescent, etc.).
>
> References
>
> [7] Lin, Y. C., & Lin, Y. H. (2017, July). A study of color illumination effect on the SNR of rPPG signals. In 2017 39th Annual International Conference of the IEEE Engineering in Medicine and Biology Society (EMBC) (pp. 4301-4304). IEEE.

---

> > ### Comment · Reviewer_Z3zg · 2021-11-25
> > **Reviewer comments**
> >
> > Adding the citation in P5 helps to better support that previous work for video-based PPG measurement are based on the first derivative.
> >
> > Good clarification of point P7 regarding lighting effects in rPPG, the paper you cite is interesting.

---

### Official Review · Reviewer_qH24 · 2021-11-02

**Correctness:** 3
**Technical Novelty And Significance:** 1
**Empirical Novelty And Significance:** 3
**Recommendation:** 5
**Confidence:** 4

**Main Review:**

Strengths:
-This is an interesting application. It is somewhat surprising how much physiological information can be extracted from simple video. This in turn brings up interesting privacy & ethical concerns which the authors briefly discuss in Section 7. I think quite a few researchers will find it interesting that so much physiological information can be extracted from a simple video of someone
-The paper is for the most part well written
-Source code will be provided. However the dataset cannot be released

Weaknesses
-Algorithmically the work is not really novel. It is a rather vanilla neural network trained on a somewhat novel dataset.
-Section 3 feels a bit detached from the rest of the paper. As far as I can tell this does not contribute to the paper's main argument.
-In section 1 the authors can provide a better description of what PPG is. Perhaps provide there a figure illustrating what type of graph data is typically provided. They may also want to consider providing up-front a diagram describing their system
-The authors are using 30fps video. How reliable are the second derivatives in such a case? If this was a high frame rate camera would they expect better results? How do intrinsic camera parameters like autogain & shutter speed affect the system reliability? How does environment illumination affect system reliability? Do the subjects need to be in a controlled environment for this to work? A discussion on this would be helpful

**Summary Of The Paper:**

This is an application paper providing an algorithm for estimating the left ventricle ejection time interval (LVET), which is the time for heart systole to occur. This metric has various clinical applications. The authors extract this metric using Photoplethysmography (PPG) labels as training labels, and video of faces as input. The authors use a DNN to extract from the images subtle changes in skin color and thus estimate blood flow changes that enable them to estimate the LVET. The authors argue that by extracting second order derivatives in the trained DNN they are able to obtain improved results

**Summary Of The Review:**

Overall as indicated above technically the algorithmic contributions are rather modest. The main novelty lies in the application described. If the paper is to be accepted it should be accepted due to the interesting application.

---

> ### Author Response · Authors · 2021-11-22
> **Reviewer Response**
>
> We want to thank the reviewer for their interest in our paper and for taking the time to give great feedback. The reviewer asks several important questions about the work, which we do our best to answer below. Additionally, the reviewer brings up a few points we hope to clarify:
>
> > P1. Algorithmically the work is not really novel
>
> We agree with the reviewer that the novelty of the proposed architecture is limited. However, this was specifically by design, as the goal of our work was not to propose a new architecture but rather to emphasize the importance of incorporating higher-order inputs/target signals into the model/optimization procedure.
>
> > P2. Section 3 feels a bit detached from the rest of the paper. As far as I can tell this does not contribute to the paper's main argument.
>
> We apologize to the reviewer if the section transition was abrupt. The goal of Section 3 was to provide a mathematical basis motivating the use of second-derivative changes in pixel intensity to be able to estimate the second-derivative changes in the PPG waveform. If there is anything else we can clarify here please let us know.
>
> > P3. In section 1 the authors can provide a better description of what PPG is. Perhaps provide there a figure illustrating what type of graph data is typically provided.
>
> Figure 1 contains a graphical depiction of a typical PPG waveform.
>
> > P4. The authors are using 30fps video. How reliable are the second derivatives in such a case? If this was a high frame rate camera would they expect better results?
>
> We are running experiments with 120fps videos and will add results as soon as we have them.
>
> > P5. How do intrinsic camera parameters like autogain & shutter speed affect the system reliability?
>
> It is known that these parameters impact remote photoplethysmography measurements. In all the datasets: AFRL, UBFC, PURE these camera settings were held constant within the dataset recordings, to avoid them impacting temporal measurements of the PPG. Studies exist on the comparison of multiple cameras running in parallel during data collection [1,2] and offer some confidence that signal recovery can be robust over widely varying imager properties.
>
> > P6. How does environment illumination affect system reliability? Do the subjects need to be in a controlled environment for this to work?
>
> Environment illumination is an important factor in system reliability for a few reasons. Firstly, if the subject is not sufficiently illuminated, then less light is reflected from the subject and captured by the camera. If the amount of reflected light is not sufficient, it can be extremely difficult to recover the PPG signal. Secondly, as described in Section 3, in this work we make the assumption that the luminance is constant (i.e. I(t) = I). This is a fair assumption to make in the case of short videos and those in which the subject can control their environment (e.g., indoors).
>
> References
>
> [1] Sun, Y., Azorin-Peris, V., Kalawsky, R., Hu, S., Papin, C., & Greenwald, S. E. (2012). Use of ambient light in remote photoplethysmographic systems: comparison between a high-performance camera and a low-cost webcam. Journal of biomedical optics, 17(3), 037005.
>
> [2] Niu, X., Han, H., Shan, S., & Chen, X. (2018, December). VIPL-HR: A multi-modal database for pulse estimation from less-constrained face video. In Asian Conference on Computer Vision (pp. 562-576). Springer, Cham.

---

> > ### Comment · Reviewer_qH24 · 2021-11-28
> > **response**
> >
> > The authors made an effort to respond to my questions. However my main hesitation remains, in that this is more of an application paper with limited technical novelty. While ICLR does accept application papers, I would feel more confident ranking the paper as above the accept threshold if this was a completely novel application.

---

### Official Review · Reviewer_me2S · 2021-11-02

**Correctness:** 3
**Technical Novelty And Significance:** 2
**Empirical Novelty And Significance:** 2
**Recommendation:** 3
**Confidence:** 3

**Main Review:**

The paper is interesting as the subject is a hot topic in the community, and the development of contactless monitoring solutions is seen as key for multiple applications, from in-hospital (pediatric ICU, patients under dialysis) or remote medical appointment (especially during the COVD pandemic)
The authors have evaluated their technique with relevant tests, such as estimation of HR and LVET. Such an evaluation technique is important, as biomedical applications need to be assessed on their final goal and authors did will not to look at MSE of estimated PPG or derivatives, but to look at the accuracy of the estimated clinical features.
I have however several concerns regarding the submission:
As denoted by the authors, the use of normalized differences for the prediction of first derivative PPG has already been suggested, the innovative aspect of the paper is therefore quite limited.
The authors do not compare their approach with other techniques from the state-of-the-art, it is therefore hard to evaluate the performance of the suggested technique.
How would the technique cope with pathological rhythms, which affect the quality and/or morphology of the PPG?
The pre-processing of the videos consists in rescaling the images into 36X36 pixels. Do the authors believe that such a low resolution is enough for real-life applications?
The use of artificial data is often key for the development of machine-learning techniques in the medical field due to the scarcity of real data. I have however some doubts about the realism of the training data, do the authors believe that avatar and generated videos are realistic enough? Does the addition of the physiological variation onto the avatar skin really reflect real and noisy data?
Given the final results, it is unclear which approach the authors are suggesting? Do they believe a specific network should be trained for each application (HR, LVET, other )


**Summary Of The Paper:**

This paper describes a novel approach of extracting information from contact-less (video-based) cardiac monitoring. The authors highlight the potential of analyzing second-derivatives of the PPG signals, especially for the estimation of Left Ventricle Ejection Time intervals.
The authors propose a deep learning approach, with multiple inputs (video, but also first and second derivatives and also a multi-task approach (first and/or second derivatives of the contact PPG)


**Summary Of The Review:**

I am not sure that the paper is innovative enough for acceptance to ICLR. The authors should also compare their approach with other state-of-the-art approaches, and evaluate on difficult real-life cases (pathological rhythms, different skin colors,…)

---

> ### Author Response · Authors · 2021-11-22
> **Reviewer Response**
>
> We thank the reviewer for their time and interest in our paper, and for posing many useful questions which we hope we can sufficiently answer. Below we detail several of the great points the reviewer brings up, and hopefully present some clarification.
>
> > P1. The use of normalized differences for the prediction of first derivative PPG has already been suggested, the innovative aspect of the paper is therefore quite limited
>
> In this paper, we propose the use of higher- (eg. second-) order derivatives to improve the estimation of higher- (eg. second-) order dynamics in the PPG waveform. Indeed, the use of normalized differences for the prediction of the first derivative PPG has already been suggested and proven to be highly effective at the estimation of summary-level metrics such as average heart rate [1]. But our results show that to obtain the best second-order derivatives it is better to optimize for second-order dynamics and use second-order inputs, something that was not necessarily intuitive from the previous work.
>
> > P2. They do not compare their approach with other techniques from the state-of-the-art, it is therefore hard to evaluate the performance of the suggested technique
>
> Since the goal of this work was not to propose and evaluate a new rPPG neural architecture, but rather to emphasize the importance of incorporating higher-order inputs/target signals into the model/optimization procedure, we did not feel that it made sense to add additional popular estimation networks into the comparison. Are there specific architectures that the reviewer felt would be appropriate?
>
> > P3. How would the technique cope with pathological rhythms, which affect the quality and/or morphology of the PPG?
>
> This is an excellent question, but unfortunately a difficult one to answer due to lack of available datasets collected from participants with pathological rhythms. We are working on collecting such data, but there do not exist any public datasets that we are aware of that contain pathological rhythms.
>
> > P4. Do the authors believe that such a low resolution is enough for real-life applications?
>
> Previous work in rPPG has shown aggressive spatial averaging (i.e., downsampling) is helpful for boosting the PPG signal-to-noise ratio [2,3,4]. The averaging helps to reduce the effect of sensor noise which is generally assumed to be Gaussian. We leverage this principle here, and as demonstrated by the testing set performance, which uses real videos collected from real human participants, we achieve good results in HR and LVET estimation.
>
> > P5. Do the authors believe that avatar and generated videos are realistic enough? Does the addition of the physiological variation onto the avatar skin really reflect real and noisy data? Given the final results, it is unclear which approach the authors are suggesting? Do they believe a specific network should be trained for each application (HR, LVET, other )
>
> While there is, of course, a generalization gap between synthetic avatar data and real data based on human participants, others have shown [5] that the simulated avatar videos are indeed realistic enough to train highly accurate models that generalize to real data with good overall performance.
>
> References
>
> [1] Chen, W., & McDuff, D. (2018). Deepphys: Video-based physiological measurement using convolutional attention networks. In Proceedings of the European Conference on Computer Vision (ECCV) (pp. 349-365).
>
> [2] Verkruysse, W., Svaasand, L. O., & Nelson, J. S. (2008). Remote plethysmographic imaging using ambient light. Optics express, 16(26), 21434-21445.
>
> [3] Poh, M. Z., McDuff, D. J., & Picard, R. W. (2010). Non-contact, automated cardiac pulse measurements using video imaging and blind source separation. Optics express, 18(10), 10762-10774.
>
> [4] Wang, W., den Brinker, A. C., Stuijk, S., & De Haan, G. (2016). Algorithmic principles of remote PPG. IEEE Transactions on Biomedical Engineering, 64(7), 1479-1491.
>
> [5] McDuff, D., Hernandez, J., Wood, E., Liu, X., & Baltrusaitis, T. (2020). Advancing Non-Contact Vital Sign Measurement using Synthetic Avatars. arXiv preprint arXiv:2010.12949.

---

### Official Review · Reviewer_eB6A · 2021-11-03

**Correctness:** 4
**Technical Novelty And Significance:** 2
**Empirical Novelty And Significance:** 2
**Recommendation:** 3
**Confidence:** 5

**Details Of Ethics Concerns:**

There are no ethical concerns in this paper.

**Main Review:**

Strength:

The investigation about higher-order of dynamics in remote cardiac measurement is inspiring. The discussion about the trade-off of low and high order dynamics indeed gives insights to future work.

As called by this paper, instead of predicting the heartbeat rate, the community should pay more attention on measuring the detailed waveform that benefits more clinical monitoring applications.

Weaknesses:

The novelty of the proposed multi-derivative architecture is limited since it is an updated version of the basic network proposed in (Chen & McDuff ,2018) with minor modifications. The 3D convolution and final RNN is not new as they have been used in other rPPG estimation works.

Since the low and high order dynamics can be a general choice for rPPG estimation, the experiments should also include other popular rPPG estimation networks.

Intuitively, the high order dynamics can also be sensitive to motion interferences, such as facial motion or facial expression. It is not clear what is the degree of the motion and illumination constraints of the physical model (Eq. 5).
The unconscious facial motion and expression are inevitable in practice. Do we need heavy pre-processing that strictly align the face region for each frame? What are the pre-processing steps in this work?
Is this one of the reasons that the network is trained on a synthetic dataset?

Also, from my understanding, the cardiac measurement is based on skin color variation, which is a kind of low-level feature. It is not clear why we need a large and diverse set of subjects to train the network.

The experiment should also include popular publicly available datasets such as PURE and UBFC-rPPG.
It is also expected to compare the proposed method with the state-of-the-art on these datasets.


**Summary Of The Paper:**

This paper investigates the effectiveness of the higher-order (second derivatives)
in the context of remote cardiac measurement. A multi-derivative architecture is designed based on the basic network of previous work (Chen & McDuff ,2018). The model is trained on the synthetic dataset (McDuff et al., 2020) and test on AFRL dataset (Estepp et al., 2014) with real person. Experimental results illustrate the effectiveness of second-order dynamics on measuring detailed cardiac signal and indicate the trade-off between low and high order dynamics of LVET error and HR error.

**Summary Of The Review:**

Considering the limited novelty of the proposed network and inadequate experiments, I think this paper does not meet the standard of this conference.

---

> ### Author Response · Authors · 2021-11-22
> **Reviewer Response**
>
> We would like to thank the reviewer for taking the time to read our paper and provide insightful feedback. The reviewer brings up several great points which we detail below. These points have led us to make several additions, which we hope have improved the paper and clarified the questions raised. Our responses to the points raised by the reviewer are detailed below:
>
> > P1. Novelty of the proposed multi-derivative architecture is limited
>
> We agree with the reviewer that the novelty of the proposed network architecture is limited. However, this was by design, as the goal of our work was not to propose a new architecture but rather to emphasize the importance of incorporating higher-order inputs/target signals into the model/optimization procedure. To that end, we felt a novel architecture might actually distract from that goal.
>
> > P2. Experiments should also include other popular rPPG estimation networks
>
> As mentioned previously, since the goal of this work was not to propose and evaluate a new rPPG neural architecture, but rather to emphasize the importance of incorporating higher-order inputs/target signals into the model/optimization procedure, we did not feel that it made sense to add additional popular estimation networks into the comparison. Are there specific architectures that the reviewer felt would be appropriate?
>
> > P3. Do we need heavy pre-processing that strictly align the face region for each frame? What are the pre-processing steps in this work? Is this one of the reasons that the network is trained on a synthetic dataset?
>
> We do not need heavy pre-processing to strictly align the face region. The attention mechanism does create a soft-attention mask which helps to upweight the face region and downweight the background. We cropped the video frames so that the face was approximately in the center and the density of skin pixels was reasonably high. The only other preprocessing was frame normalization and differencing. The pre-processing steps in this work are described in the paper, but we will make an effort to draw them out more clearly.
>
> > P4. It is not clear why we need a large and diverse set of subjects to train the network.
>
> In machine learning, a large number of samples with sufficient representative variation is required to properly learn parameters that minimize training error and ideally maximize generalization to unseen, future samples. The diversity of the training and testing populations is quite important for this particular problem because light is absorbed and reflected differently for different skin tones [1]. Prior work in the rPPG domain specifically has shown that similar biases exist when using non-contact monitoring [2].  Therefore, leveraging a large and diverse dataset is necessary for achieving sufficient prediction performance across many different populations. We have updated the paper to better clarify this motivation.
>
> > P5. The experiment should also include popular publicly available datasets such as PURE and UBFC-rPPG. It is also expected to compare the proposed method with the state-of-the-art on these datasets.
>
> We thank the reviewer for this point. While we did evaluate on a separate dataset containing videos of real humans, this was indeed only a single dataset for measuring testing performance. Therefore, we have generated and reported results for the PURE and UBFC-rPPG datasets for the FD-optimized model and the SD-optimized model (Appendix Table 3). When testing both the FD-optimized and SD-optimized models on the PURE and UBFC datasets, we see the same trend that was reported in the AFRL dataset: the FD-optimized model achieves a lower MAE when estimating HR compared to the SD-optimized model, but the SD-optimized model sees a significant improvement in the LVET MAE compared to the FD-optimized model. These new results provide additional evidence that the trend we observed in the AFRL dataset is likely to hold in general.
>
>
> References
>
> [1] Bent et al. Investigating sources of inaccuracy in wearable optical heart rate sensors; npj Digital Medicine (2020).
>
> [2] Dasari et al. Evaluation of biases in remote photoplethysmography methods; npj Digital Medicine (2021).

---

> > ### Comment · Reviewer_eB6A · 2021-12-02
> > **Reply to : Are there specific architectures that the reviewer felt would be appropriate?**
> >
> > I understand this work is to address the importance of incorporating higher-order inputs/target signals.
> > To this end, the experiments should evaluate the effect of incorporating higher-order inputs/target signals
> > in different types of rPPG estimation networks (at least the popular ones), such as the rPPGNet (without the STVEN module) in [1], DeeprPPG [2] which use local ROIs for estimation, and methods use STMap as input such as RhythmNet [3] and CVD [4].
> >
> > This paper only evaluates the method described in Fig. 2, which cannot demonstrate whether incorporating higher-order inputs/target signals is a generalized cue that can be applied in any rPPG estimation networks, or, what are the requirements of the network of using this cue.
> >
> > [1] Yu et.al., Remote Heart Rate Measurement from Highly Compressed Facial Videos: an End-to-end Deep Learning Solution with Video Enhancement. ICCV 2020
> >
> > [2] Liu et.al., A General Remote Photoplethysmography Estimator with Spatiotemporal Convolutional Network. FG 2020
> >
> > [3] Niu et.al., RhythmNet: End-to-end Heart Rate Estimation from Face via Spatial-temporal Representation. TIP 2020
> >
> > [4] Niu et.al., Video-based remote physiological measurement via cross-verified feature disentangling. ECCV 2020

---

### Decision · Program_Chairs · 2022-01-20

**Decision:**

Reject

**Comment:**

The paper has received 5 reviews with 4 advocating for rejection (marginal or clear cut) and one borderline leaning towards a weak accept. The key concerns voiced by the reviewers are the lack of novelty (*the novelty of the proposed multi-derivative architecture is limited*), the lack of comparisons with specific architectures in appropriate setting (rPPGNet without the STVEN module, DeeprPPG, RhythmNet, CVD), and concerns about the use of synthetic data (although authors provide some justifications to that end). It appears that the key to reviewers' scores is that higher-order dynamics did not constitute a sufficient novelty.

Given the post-rebuttal scores and discussions, AC has no option but to recommend a reject at this point.